# Definition of a Protocol to Manage and Officially Confirm SHB Presence in Sentinel Honeybee Colonies

Giovanni Formato [1], Giovanni Federico [2], Camilla Di Ruggiero [1], Marco Pietropaoli [1,*], Marcella Milito [1] and Franco Mutinelli [3]

1   Istituto Zooprofilattico Sperimentale del Lazio e della Toscana "M. Aleandri", 00178 Rome, Italy; giovanni.formato@izslt.it (G.F.); diruggierocamilla@gmail.com (C.D.R.); marcella.milito@izslt.it (M.M.)
2   Istituto Zooprofilattico Sperimentale del Mezzogiorno, Loc. Catona, 89135 Reggio Calabria, Italy; giovanni.federico@izsmportici.it
3   Istituto Zooprofilattico Sperimentale delle Venezie, NRL for Honey Bee Health, 35020 Legnaro, PD, Italy; fmutinelli@izsvenezie.it
*   Correspondence: marco.pietropaoli@izslt.it

**Abstract:** Given the consolidated circulation of *Aethina tumida* (SHB) in Reggio Calabria and Vibo Valentia provinces of Calabria region (Southern Italy), the need for a more effective and less time-consuming approach to SHB surveillance emerged. Accordingly, honeybee sentinel colonies were established in the infested areas under the supervision and management of the Veterinary Services of the Local Health Unit. In this short communication, we present the protocol adopted in the Calabria region to manage the SHB positive sentinel honeybee colonies. The procedures for safely packing and transport the SHB infested sentinel honeybee colonies from the field to the official laboratory and the subsequent procedure for their careful inspection in the laboratory are illustrated.

**Keywords:** *Aethina tumida*; beehive; inspection; honeybee; sentinel colony

## 1. Introduction

The Small Hive Beetle (SHB), *Aethina tumida* Murray (Coleoptera: Nitidulidae), is an endemic parasite of sub-Saharan Africa, which affects honeybees. The beetle can cause damages to the colonies, e.g., larvae that burrow and tunnel through the comb, piercing and damaging the wax comb and cell caps, eat honey, pollen and live honeybee brood (eggs, larvae and pupae), honeybee queens may stop egg laying and the colony may quickly weaken, the honey bee colony may leave when SHB infestation is heavy and colony loss can also occur. Furthermore, larvae defecate in the honey causing it to ferment, froth and weep from the cells within the hive or in the supers stored in honey houses. Contaminated honey is unsuitable for sale and unacceptable to bees as bee food [1]. During the last 25 years, SHB was introduced into many countries worldwide [1] through the exchange and import of bees and beekeeping equipment and through transhumance [2]. In September 2014, the presence of SHB was officially confirmed in Calabria and, some months later, in Sicily regions (Southern Italy) [3]. The infestation with SHB is a notifiable disease in the EU [4,5] and the Italian Ministry of Health promptly set up a surveillance and eradication/containment plan [6]. Eradication measures are based on the thorough clinical inspection of the beehives of the apiaries in the concerned area [7–9], including the stamping out of the SHB infested colonies on site. Early detection in a surveillance plan is crucial, even more, when eradication is the goal [10].

Further to September 2014, the presence of *A. tumida* in the provinces of Reggio Calabria and Vibo Valentia in the region of Calabria has been demonstrated in the following years and is still circulating in that territory based on the surveillance activity carried out by the Veterinary Services of the Local Health Unit (National Health Service) and regularly reported to the central authority [11].

A protection zone of a 20 km radius around the first place of *A. tumida* detection was firstly established and progressively extended to the whole territory of the two provinces since the number of reports increased and the affected territory expanded [3]. Furthermore, the inspective activity of the Veterinary Services of the Local Health Unit decreased and the need for a different surveillance approach emerged.

In 2015, when the SHB infestation appeared well consolidated in the above-mentioned provinces, the use of sentinel colonies [12,13] within and at the borders of the protection zone was introduced among the control measures adopted to collect more information about the presence and spreading of the infestation (Figure 1).

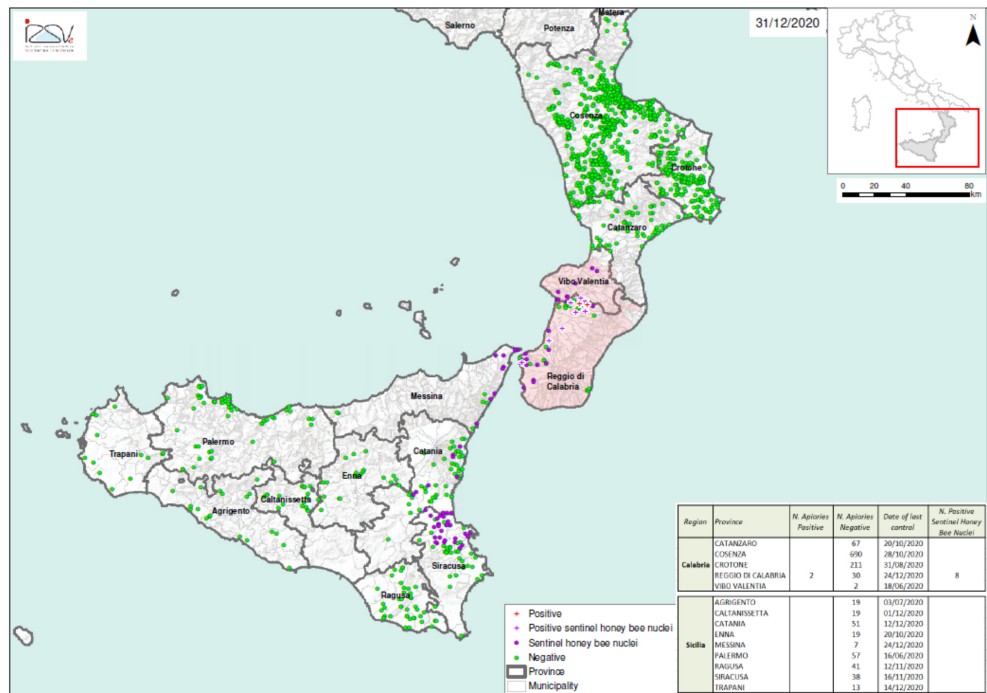

| Region | Province | N. Apiaries Positive | N. Apiaries Negative | Date of last control | N. Positive Sentinel Honey Bee Nuclei |
|---|---|---|---|---|---|
| Calabria | CATANZARO | | 67 | 20/10/2020 | |
| | COSENZA | | 690 | 28/10/2020 | |
| | CROTONE | | 211 | 31/08/2020 | |
| | REGGIO DI CALABRIA | 2 | 30 | 24/12/2020 | 8 |
| | VIBO VALENTIA | | 2 | 18/06/2020 | |
| Sicilia | AGRIGENTO | | 19 | 03/07/2020 | |
| | CALTANISSETTA | | 19 | 01/12/2020 | |
| | CATANIA | | 51 | 12/12/2020 | |
| | ENNA | | 19 | 20/10/2020 | |
| | MESSINA | | 7 | 24/12/2020 | |
| | PALERMO | | 57 | 16/06/2020 | |
| | RAGUSA | | 41 | 12/11/2020 | |
| | SIRACUSA | | 38 | 16/11/2020 | |
| | TRAPANI | | 13 | 14/12/2020 | |

**Figure 1.** Map showing the location of sentinel honeybee colonies and colonies positive and negative to small hive beetle in Calabria and Sicily regions.

A sentinel honeybee colony is composed of 3 frames of brood covered with honeybees and two frames with honey and pollen and the queen placed in a hive intended for artificial swarms or nuclei. Sentinel colonies are created with the help of beekeepers and are taken from territories still considered free from SHB. While moving these colonies to their dedicated location they are kept close and protected with a net with a mesh of 2 mm.

A pair of sentinel honeybee colonies were used in each surveillance site (Figure 2) that covered an area of 2660 m side (approx. 7000 km$^2$).

They are under the supervision of the Veterinary Services of the Local Health Unit (National Health Service) and are directly managed by them. The sentinel is managed almost like a regular colony. In the beginning, orphan colonies were established but it appeared very soon that their survival was at risk; then, the decision was taken to use only queenright sentinel colonies. The sentinel is supposed to be fully inspected about every 20 days for SHB and new queen cells are systematically removed to prevent swarming. If the colony gets too strong, brood frames are removed, and foundations can be added. Supers are not added. Traps are applied within the hive or on the bottom board to favor SHB capture.

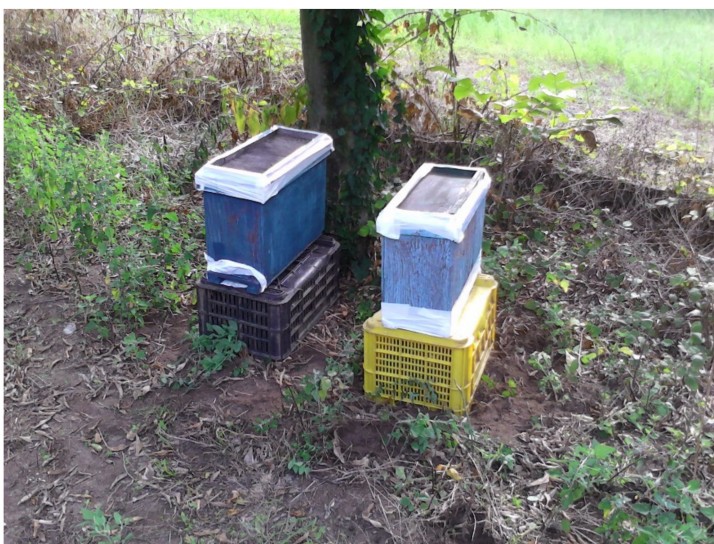

**Figure 2.** A pair of sentinel honey bee colonies sealed following SHB detection.

A field inspection of sentinel colonies does not differ very much from the one a beekeeper would perform in his colonies, i.e., remove the lid and inspect the inner side, remove the inner cover, inspect it and check the top of the hive, then remove each frame and carefully inspect them one by one, also inspect the sides and the bottom board of the hive. As soon as a beetle is detected, the colony is immediately closed and sealed to avoid or at least limit any dispersal of SHB. Then secure packaged and sent to the official laboratory for further examination.

The advantages of using sentinels are that their inspection is less time-consuming than a full-size colony, they are efficient in attracting and revealing the presence of SHB, they are cheaper, easily manageable and placed outside the apiaries so that the Veterinary Services can freely access them at any time. In addition, the competent authority can easily carry out the procedure of killing and packing under biosafety conditions when the sentinel is found to be infested by SHB as well.

This was considered a reliable tool to quickly report the spread of the beetle outside the protection zone. Since then, the use of honeybee sentinel colonies are still in force today as part of the national SHB surveillance program [14] and is currently a practice strongly recommended to facilitate and possibly anticipate the detection of SHB.

The same tool was adopted in and around the protection zone established in the Sicily region both in 2014 and 2019 following the detection of SHB infested colonies. In addition, sentinel colonies were established in the Sicily region along the Strait of Messina facing the coast of the Calabria region as it was considered an area at risk of introduction of SHB.

The use of sentinel sites for the detection of invasive honeybee pests and diseases was proposed in the UK by Keeling et al. [15]. In this case, sentinel sites with apiaries were established in areas at risk for the introduction of pests and diseases. In the case herein described (Calabria and Sicily regions), the use of sentinel colonies was decided to improve the detection of SHB in the surveillance program established in the infested areas of the southern Italian territory.

In this short communication, we present the protocol adopted in the Calabria region (Southern Italy) to manage the SHB positive sentinel colonies. The procedures for packing and transportation of the SHB infested sentinel colonies from the field to the official laboratory, and the subsequent procedure for their inspection in the laboratory are illustrated.

## 2. Materials and Methods

According to Regulation (EU) 2016/429 [16], once the Veterinary Services detect the presence of the SHB in the sentinel colonies, they capture or collect adults and/or larvae

and kill them immediately by immersing them in a tube containing 10–15 mL of 70% ethanol. The Veterinary Services send the collected beetles together with the sentinel colonies to the official laboratory for identification, according to the biosecurity measures in force at the national level:

- Seal all the cracks of the hive with adhesive tape;
- Kill the bees using a sulfur dioxide cylinder [17] and administered by a pipe through the entrance of the hive;
- Wrap up the hive completely, using multiple layers of polyethylene film for packaging (Figure 3);
- Seal the packaging with adhesive tape (Figure 4);
- Introduce the packed hive inside four heavy plastic bags and, before introducing each bag into the next, seal it with the adhesive tape (Figure 5);
- Transport it with no delay to the official laboratory or store it at −18 °C until shipping.

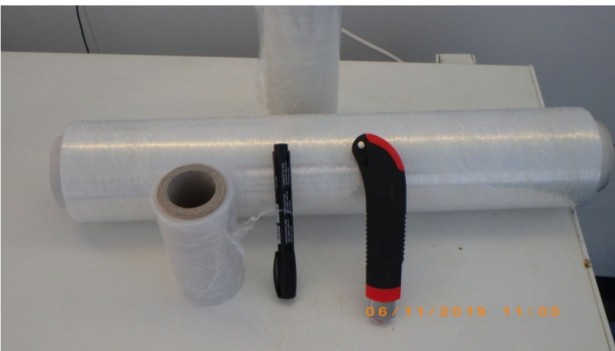

**Figure 3.** Polyethylene film for packaging.

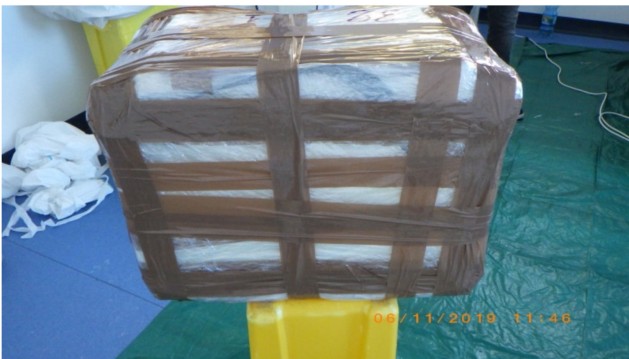

**Figure 4.** Seal the packaged sentinel beehive with adhesive tape.

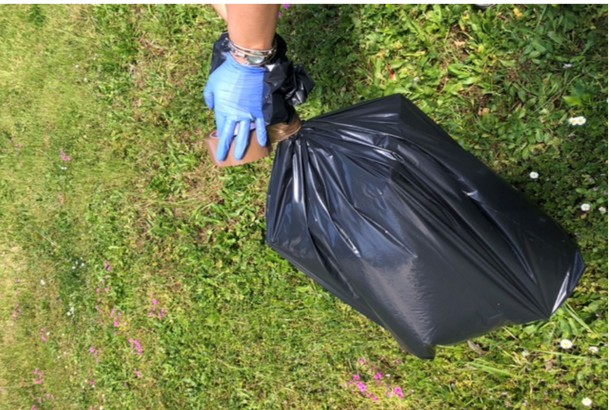

**Figure 5.** Sentinel colony inside heavy plastic bags.

- Store the packed sentinel in the official laboratory at −12 °C or less (core temperature) for at least 24 h [18] before proceeding with its visual inspection;
- Remove the four heavy plastic bags one by one and carefully inspect them for the presence of beetles;
- Cut and remove the polyethylene film and look for the presence of the parasite or its parts with the help of a magnifying glass. Carefully inspect the adhesive tape (Figure 6) and the cracks of the hive since SHBs could have been trapped by the adhesive tape's glue;
- Once the packaging is removed, open the lid of the hive and inspect it for the parasites with a magnifying glass and under an appropriate source of lighting;
- Remove the honeybees on the lid by dropping them inside the hive;
- Shake all the bees inside the hive from the top of the frames and from the hive edges;
- Extract and clean the frames from the honeybees one by one, dropping them inside the hive and placing the frames on a sheet of white paper for inspection;
- Carefully inspect all cells of the frames looking for *A. tumida* larval or egg forms.;
- Remove honeybees from inside the cells to look for the presence of the parasite under them (Figure 7) and perform the same operation on all frames.
- Look for *A. tumida* on the hive walls and on spacers;
- Accumulate the honeybees in the lower part of the hive and drop about 300 honey bees in a tray covered with a sheet of white paper;
- Look for the presence of beetles among the honeybees, with the help of a magnifying glass (Figure 8) and under adequate lighting, and separate them with a tweezer;
- Repeat the same operation to examine all the honeybees in the hive;
- At the end of the inspection, put all the material in a clean container for later storage or disposal by incineration;
- If SHB is detected, remove it with a tweezer and place it in a Petri dish for further identification under the stereomicroscope in order to differentiate it from morphologically similar coleopterans.

Between December 2014 and January 2015, a preliminary investigation has been carried on other Nitidulidae beetle species occurring on rotten fruit in the Reggio Calabria province infested with small hive beetle [19]. A total of 78 adult beetles belonging to eight different species of the Nitidulidae family were detected on rotten citrus fruit. The beetles collected were characterized by the clubbed antennae typical of Nitidulidae family, but in general by thin elongated bodies that macroscopically differed from the much broader *A. tumida* [1]. Similar results have been obtained also in further field surveys carried out in the following years until 2020 (Mutinelli, personal communication).

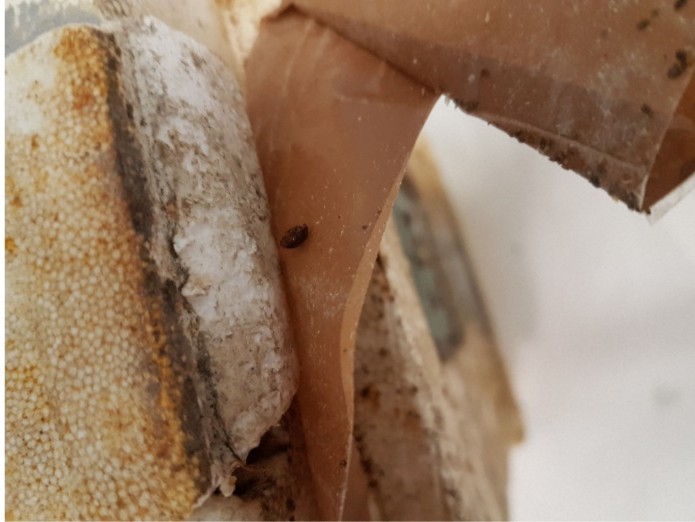

**Figure 6.** Seek the beetle on the adhesive tape used to seal the package.

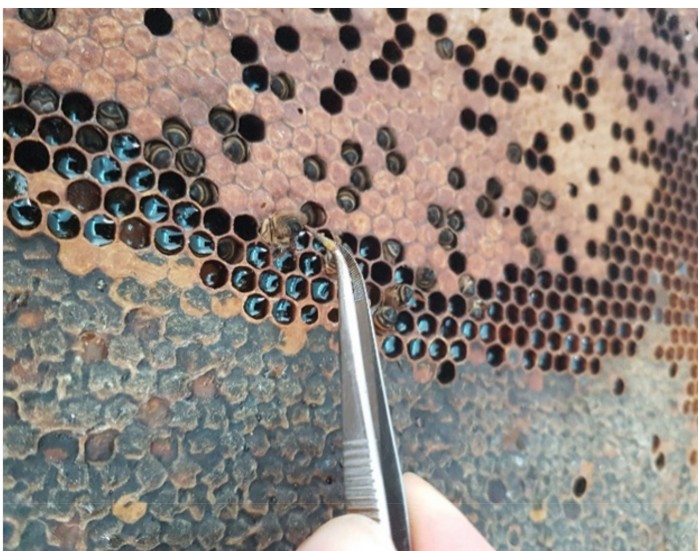

**Figure 7.** Removal of the honeybees from the cells with a tweezer.

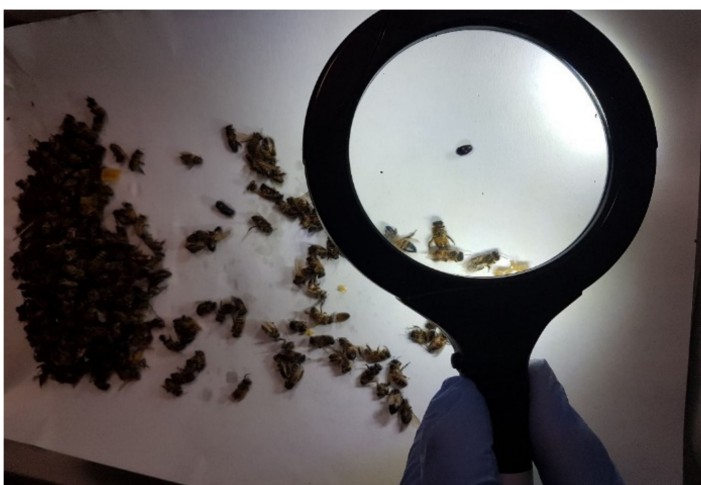

**Figure 8.** Collected honeybees are screened for beetles with the help of a magnifying glass.

### 3. Discussion

In the case herein described sentinel honeybee colonies have been established to ease and improve the detection of SHB in the framework of the Italian national surveillance program applied in the infested areas of Calabria and Sicily regions. In each monitoring site, a pair of sentinel honeybee colonies are established under the supervision of the Veterinary Services of the Local Health Unit (National Health Service) and directly managed by official veterinarians.

The advantages of relying on sentinels for SHB surveillance are as follows: inspection is less time consuming than a full-size colony, easily manageable by an official veterinarian, efficient in attracting and revealing the presence of SHB, cheaper (according to the Italian National Agriculture Bulletin the estimated cost is about 10,000 euro) than a full-size colony (they are killed and destroyed if found positive), and placed outside the apiaries so that Veterinary Services can freely access to them at any time. In addition, the competent authority can easily carry out the procedure of sanitary restriction, killing and packing under biosafety conditions when the sentinel is found infested by SHB.

Additionally, disadvantages do exist, e.g., the sentinel does not always result as attractive to SHB as full colonies, SHB detection could depend also on the density of honey bee colonies in a certain area, specific training of official veterinarians is needed for both hive inspection and hive management, the number of official veterinarians trained and

available for the management of sentinels is not always adequate, sometimes the support of beekeepers is required. Sensitization of both veterinarians and beekeepers on the reliability of sentinel colonies for a SHB surveillance program is very important and can strongly improve the success of the intervention. The Sicily region could be taken as an example of this collaboration. In fact, they have been able to detect and eradicate SHB twice following incursions in 2014 and 2019.

We are confident that a sentinel system is capable to detect SHB more easily than inspection of full colonies in apiaries and could efficiently contribute to the surveillance program that is fundamental to break down the progress of SHB in Italy. Of course, once established, the sentinel system requires to be maintained active for long periods, probably years.

Furthermore, the development and implementation of this type of approach should be recommended to not yet infested areas that could be at risk of invasion.

In addition to what was already mentioned, it should be clear that when designing a similar monitoring system, an appropriate knowledge of the territory is needed in terms of honeybee colony population, beekeepers and the beekeeping industry.

Attention has also been devoted to managing SHB infested sentinel colonies in order to prevent any escape of beetles, i.e., sealing any opening of the hive, the quick and safe killing of the colony, biosecure packing with multiple layers of polyethylene film, adhesive tape, multiple plastic bags and transportation with no delay to the official laboratory or storage at $-18\ °C$ until shipping. The use of multiple layers of polyethylene film was suggested also by Mutinelli et al. [20] to seal supers full of honey when transporting them from SHB non-infested, but at-risk areas, to extraction and processing facilities located in SHB non-infested areas.

In conclusion, the procedures for packing and transport SHB infested sentinel honeybee colonies from the field to the official laboratory, and the subsequent protocol for their careful inspection in laboratory conditions are considered easy to apply, useful on a routine application and able to grant the required biosafety conditions.

**Author Contributions:** Conceptualization, G.F. (Giovanni Formato) and M.P.; methodology, G.F. (Giovanni Formato) and G.F. (Giovanni Federico); investigation, G.F. (Giovanni Formato), M.P. and G.F. (Giovanni Federico); writing—original draft preparation, C.D.R., M.P., M.M., G.F. (Giovanni Formato) and G.F. (Giovanni Federico); writing—review and editing, F.M. All authors have read and agreed to the published version of the manuscript.

**Funding:** This research received no external funding.

**Institutional Review Board Statement:** Not applicable.

**Informed Consent Statement:** Not applicable.

**Data Availability Statement:** Not applicable.

**Acknowledgments:** Authors would like to thank Veterinary Services of the Local Health Unit of Reggio Calabria province involved in the official control activities for SHB.

**Conflicts of Interest:** The authors declare no conflict of interest.

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
