# Peer review of "Definition of a Protocol to Manage and Officially Confirm SHB Presence in Sentinel Honeybee Colonies"

_applsci, doi:10.3390/app11178260_

Round 1

Reviewer 1 Report

Review of : Formato, G.; Federico, G.; Di Ruggiero, C.; Pietropaoli, M.; Milito, M. Mutinelli, F. Definition of a protocol to manage and officially confirm SHB presence in sentinel honey bee colonies . Appl. Sci

Comments (mainly general issues):

In their communication, Formato et al. present a protocol employed in Italy to inspect honey bee colonies used as sentinels for the detection of Small Hive Beetle (Aethina tumida, SHB). SHB is a serious and expanding threat to the honey bee, so the topic presented here is of major importance.

The article is in general well written.

This article however needs more work before publication:

1/ Many points are missing: management of sentinel honey bee colonies is completely missing in this article. It would have been a great opportunity for the reader to learn more about the practical use of this kind of setup based on the Italian experience. Without this information, information of management of sentinel hives after SHB detection is of poor use.

I highly recommend to add a part on this topic containing following information:

-map showing current location of sentinel honey bee colonies,

-how are they concretely managed? Are measures taken to prevent swarming? What is done if colony gets too strong? Are supers added? If yes, what is done with the honey/who is in charge of extracting it? Are measures taken to avoid SHB spread during this step?

-How does a field inspection of the sentinel colonies take place? What special care is given to not inadvertently move SHB? In what extend are colony inspections different from those which a beekeeper would perform himself?

-How are these sentinel colonies created? What measures are taken to prevent SHB migration while moving these colonies to their dedicated location?

2/ Material and methods is a list of actions to undertake to inspect the killed presumably SHB-positive colonies. This part is difficult to read: could you transform it to something more reader-friendly? For instance by means of a figure? It would make your work more easy to be directly used in the field if you would provide a kind of step-by-step checklist or a kind of linear timeline. In addition, this part is first written in passive form (‘’is cut’’, ‘’are inspected’’) and then there is a sudden switch at L 106 to something more directive (‘’Open the lid’’, ‘’inspect’’). Please choose one kind of expression throughout this part to improve consistency.

3/ Discussion is more or less a copy-paste of introduction and material and methods. This doesn’t bring much to the reader. Please use this part of your article to discuss more about the relevance/the advantages/the disadvantages of your protocol. How to improve weaknesses? Are you confident in the capacity of the sentinel colonies to break down the progress of SHB in Italy? Would you recommend to others to develop this type of approach? What would be the main points, including practical issues, to take in mind while designing a comparable monitoring system? Including some information about the cost of the system may be highly interesting, as it could limit exporting the approach to countries with less funding capacities.

Reviewer 2 Report

This short communication by Formato et al. provides a useful information on a method that can be used for early detection of small hive beetle parasite. Given the worldwide spread of this parasite, this paper should be of high interest to many readers. However, there are some minor comments and suggestion that can help the authors to improve their paper. Please see them attached. 

Round 2

Reviewer 1 Report

The manuscript by Formato et al. has been greatly improved since the last version and I thank the Authors for their work. Currently all needed information to enable a good understanding of the protocol and sentinel colony dissection are present. In order to prepare the manuscript for publishing, I would however recommend to have the paper proofread by a professionnal English editing service, as some sentences are either too long or do not sound very English. I however have no qualification to propose improvements myself and let the decision to the Editor.